# Factors associated with gynaecological morbidities and treatment-seeking behaviour among adolescent girls residing in Bihar and Uttar Pradesh, India

Pradeep Kumar[1], Shobhit Srivastava[1], Shekhar Chauhan[2], Ratna Patel[3], Strong P. Marbaniang[3]*, Preeti Dhillon[1]

1 Department of Mathematical Demography & Statistics, International Institute for Population Sciences, Mumbai, India, 2 Department of Population Policies and Programmes, International Institute for Population Sciences, Mumbai, India, 3 Department of Public Health and Mortality Studies, International Institute for Population Sciences, Mumbai, India

* marbaniangstrong@gmail.com

## Abstract

### Background

Gynaecological morbidities are more common than reproductive and contraceptive morbidities and constitute a substantial proportion of disease burden in women. This study aimed to examine the prevalence and factors associated with gynaecological morbidities and the treatment-seeking behaviour among adolescent girls residing in Bihar and Uttar Pradesh, India.

### Methodology

The study utilized data from the Understanding the Lives of Adolescents and Young Adults (UDAYA) survey with a sample size of 14,625 adolescents girls aged 10–19 years. We defined gynaecological morbidity in dichotomous form, created from five questions on different morbidities. Further, the treatment-seeking behaviour was assessed for reported gynaecological morbidities three months prior to the survey. Univariate and bivariate analysis was used to perform analysis to carve out the preliminary results. Additionally, the study employed the heckprobit selection model, a two-equation model, to identify the determinants of outcome variables.

### Results

Overall, about one-fourth (23.6%) of the adolescent girls reported suffering from gynaecological morbidities, and only one-third of them went for treatment. Non-Scheduled Caste/Scheduled Tribe (Non-SC/ST) adolescents were significantly less likely to have gynaecological morbidities (β: -0.12; CI: -0.18, -0.06) compared to SC/ST counterparts; however, they were more likely to go for the treatment (β: 0.09; CI: 0.00, 0.19). The adolescents who had 8–9 (β: 0.17; CI: 0.05, 0.29) or ten and above years of education (β: 0.21; CI: 0.09, 0.34)

**Data Availability Statement:** There are restrictions on sharing the data, which was collected by Population Council, India and ethically approved by

ethical committee of Population Council, India. The data are now owned and stored by Harvard Dataverse, and other researchers may submit data access requests via the following URL: https://dataverse.harvard.edu/dataset.xhtml?persistentId=doi:10.7910/DVN/RRXQNT.

**Funding:** This paper was written using data collected as part of Population Council's UDAYA study, which is funded by the Bill and Melinda Gates Foundation and the David and Lucile Packard Foundation. No additional funds were received for the preparation of the paper.

**Competing interests:** The authors have declared that no competing interests exist.

had a higher likelihood of going for the treatment than adolescents with no education. Moreover, adolescents who belonged to rural areas were less likely to go for the treatment of gynaecological morbidities (β: -0.09; CI: -0.17, -0.01) than urban counterparts.

## Conclusion

Multi-pronged interventions are the need of the hour to raise awareness about the health-care-seeking behaviour for gynaecological morbidities, especially in rural areas. Adolescent girls shall be prioritized as they may lack the knowledge for gynaecological morbidities, and such morbidities may go unnoticed for years. Mobile clinics may be used to disseminate appropriate knowledge among adolescents and screen asymptomatic adolescents for any possible gynaecological morbidity.

## Introduction

Adolescence is a transition period of physical and psychological change from puberty to legal adulthood. Adolescent includes individuals between the ages of 10–19 years [1]. Globally more than 1.2 billion are adolescents, meaning that one in every six persons is an adolescent. In absolute numbers (243 million), India is home to more adolescents than any other country [2]. WHO defined reproductive morbidity as consisting of three types of morbidity: obstetric, contraceptive, and gynaecological; gynaecological morbidity includes any condition, disease, or dysfunction of the reproductive system which is not related to pregnancy, abortion, or childbirth but may be related to sexual behaviour [3]. Some of the gynecological morbidity symptoms include irregular menstrual patterns, white vaginal discharge, itching of vulva, burning urination, and inguinal swelling [4]. Globally gynaecological problems are the significant contributors to morbidity and mortality, with the highest burden of disease borne by women in the low resource countries. The gynaecological disease is attributed to approximately 4.5 percent of the overall global disease burden, which exceeds that of other major global diseases such as malaria, tuberculosis, ischaemic heart disease, and maternal conditions [5].

Menstruation is often traumatic and very negative experience for young girls in most parts of India. Many traditional beliefs, misconceptions, and practices are associated with menstruation, which makes girls vulnerable to stress and depression as well as reproduction problems [6]. Evidence from India's existing studies shows that a large proportion of girls suffers from various gynaecological morbidity [6]. The population-based cross-sectional study reveals that 15 percentage of Indian adolescent girl suffers from any form of gynaecological morbidity and the prevalence varies by socio-demographic characteristics [7]. Heavy menstrual bleeding, dysmenorrhea, menstrual irregularities, primary and secondary amenorrhea are common gynaecological problems among adolescent girls. A study in Maharashtra and Bangladesh reported that menstrual disorder, dysmenorrhea, and prevaginal discharge, and vulval itching as the common gynaecological problem among adolescent girls [8, 9]. Despite being a common problem during puberty and adolescence, they also run the risk of delayed diagnosis and treatment [10]. The youth survey from six Indian states reported low treatment-seeking for symptoms of reproductive tract infections (RTI's) by married and unmarried young women (15–24 years), and the factor such as stigma, shame, and social isolation are more likely to deter unmarried youth from seeking treatment for RTI's [11]. Studies indicate that delayed seeking treatment is because most adolescents did not seriously concern their reproductive health

problems or pain but only sought treatment when the pain was unbearable [12]. Another study among adolescents from Bangladesh mentioned that the reason for not receiving treatment for gynaecological problems includes lack of knowledge, economic hardship, shyness to expose to doctor, and no need for treatment for the problems [13]. Seeking treatment for gynaecological morbidity by adolescents is a complex process. It mainly depends on the individual's comfort and familiarity with the service providers and the accessibility to the health services [14].

Despite the significant proportion of the adolescent population in India, studies have highlighted a lack of information on adolescents' sexual and reproductive health [15]. Existing studies have indicated that programs and policies on sexual and reproductive health should give special attention to young and adolescent girls in India [16, 17]. Only a few studies in India have focused on adolescents' gynecological morbidity and their treatment-seeking [11, 18]. This paper contributes to the literature on the prevalence and treatment-seeking behaviour of gynaecological morbidity with a particular focus on adolescent girls. In our analysis, we apply the Heckman model approach to explore the socio-economic determinants of treatment-seeking behaviour. The advantage of using the Heckman model approach is that it improves the estimates by accounting for the unobserved or unmeasured factors that may influence both the outcome (seeking treatment) and the selection (having any gynaecological disease) variable [19, 20]. The objective of the study is to determine the factors associated with gynaecological morbidity and treatment-seeking behaviour among adolescents in Bihar and Uttar Pradesh.

## Methods

### Data

The authors used secondary source of data collected by Population Council, New Delhi, India. The Population Council Institutional Review Board provided ethical approval for the study. Adolescents provided individual written consent to participate in the study, along with a parent/guardian for unmarried adolescents younger than 18 years. The study utilized data from the Understanding the Lives of Adolescents and Young Adults (UDAYA) project survey conducted in two Indian states Uttar Pradesh and Bihar, in 2016 by Population Council under the guidance of the Ministry of Health and Family Welfare, Government of India [21]. The survey collected detailed information on family, media, community environment, assets acquired in adolescence, and quality of transitions to young adulthood indicators. The sample size for Uttar Pradesh and Bihar was 10,350 and 10,350 adolescents aged 10–19 years, respectively. The required sample for each sub-group of adolescents was determined at 920 younger boys, 2,350 older boys, 630 younger girls, 3,750 older girls, and 2,700 married girls in both states. The effective sample size for this study was 14,625 adolescents girls aged 10–19 years. The UDAYA adopted a multi-stage systematic sampling design to provide the estimates for states as a whole as well as urban and rural areas of the states [21].

### Outcome variables

The explanatory variable was formed using the following questions that a) Have you had experienced genital ulcers in the last three months? b) Have you had experienced itching in the genitals in the last three months? c) Have you had experienced swelling in the groin in the last three months? d) Have you had experienced burning while passing urine in the last three months? e) Have you had experienced white discharge in the last three months? The response of the questions was coded as 0 means "no," and 1 means "yes." Now the variable named gynaecological morbidity was generated using the above five questions. If the respondent had

experienced any issue from the questions mentioned above, then it was coded as 1 means "yes," and if the respondent had experienced none of the above, it was coded as 0 means "no." Apart from this, treatment-seeking behaviour was assessed using the question "Did you seek treatment for this complaint?" the response was coded as 0 "no" and 1 "yes." Hence both the outcome variables were binary.

## Predictor variables

The predictor variables were selected after going through the extensive literature review [4, 6, 7, 11].

## Individual variables

1. Sexually active variable was generated using "whether the respondent was married or not?" and "whether or not she had sexual intercourse with her boyfriend? ", if the response was yes in either of the cases, then she was coded as sexually active 1 "yes" and in the other case as sexually inactive 0 "no."

2. Use of sanitary napkin was coded as "sanitary napkin," "cloth," and "others."

3. Toilet facility was coded as "own flush/pit," "shared flush/pit," and "no facility."

4. Age was coded as 10–12, 13–14, 15–17, and 18–19 years.

5. Education was coded as "no education," "1–7 years", "8–9 years," and "10 and above years".

6. Working status was coded as "not employed" and "employed."

7. Media exposure was coded as "no," "rare," and "frequent."

## Household variables

1. Wealth index was coded as the "poorest," "poorer," middle," richer," and richest." The variable of wealth status was created using the information given in the survey. Households were given scores based on the number and kinds of consumer goods they own, ranging from a television to a bicycle or car, and housing characteristics such as the source of drinking water, toilet facilities, and flooring materials. These scores are derived using principal component analysis. Wealth quintiles were compiled by assigning the household score to each usual (de jure) household member, ranking each person in the household population by their score, and then dividing the distribution into five equal categories, each with 20 percent of the population.

2. Caste was coded as "Scheduled Caste/Scheduled Tribe (SC/ST)" and "non-SC/ST." The Scheduled Caste includes "untouchables," a group of the population that is segregated socially and financially/economically by their low status as per Hindu caste hierarchy. The Scheduled Castes (SCs) and Scheduled Tribes (STs) are among India's most disadvantaged socio-economic groups. The OBC is the group of people identified as "educationally, economically, and socially backward." The OBC's are considered low in the traditional caste hierarchy but are not considered untouchable [22].

3. Religion was coded as "Hindu" and "non-Hindu."

4. Residence was available in the data as "urban" and "rural."

5. The survey was conducted in two states, "Uttar Pradesh" and "Bihar."

## Statistical analysis

Univariate and bivariate analysis was used to perform analysis to carve out the preliminary. Additionally, the study employed the heckprobit selection model, which is a two-equation model. First, there is a selection model (in this study, referring to "Do the respondent had any gynaecological morbidities in the last three months? (Yes or no)"). Secondly, there is an outcome model with a binary outcome (in this study refers to "did the respondent went for seeking its treatment? (Yes or no)"). The model provides a two-step analysis and deals with the zero-sample issue. It can accommodate the heterogeneity (i.e., shared unobserved factors) between respondents and then address the endogeneity (between occurrence gynaecological morbidity and opting for its treatment) among adolescents. The Heckman model is identified when the same independent variables in the selection equation appear in the outcome equation [23]. However, this does not provide precise estimates in the outcome equation because of high multicollinearity; it was suggested to have at least one independent variable in the selection equation and not in the outcome equation. A p-value of less than 0.05 was considered statistically significant.

The probit model with sample selection assumes that there exists an underlying relationship:

$$y_j = x_j\beta + u_{1j} \qquad \text{latent equation}$$

such that we observe only the binary outcome

$$y_i^{probit} = (y_j > 0) \qquad \text{probit equation}$$

The dependent variable, however, is not always observed. Instead, the dependent variable for observation j is observed if:

$$y_i^{select} = (z_j\gamma + u_{2j} > 0) \qquad \text{selection equation}$$

*Where,*

$$u_1 \sim N(0, 1)$$

$$u_2 \sim N(0, 1)$$

$$Corr(u_1, u_2) = \rho$$

When $\rho \neq 0$, standard probit techniques applied to the first equation yield biased results. Heckprobit provides consistent, asymptotically efficient estimates for all the parameters in such models. For the model to be well identified, the selection equation should have at least one variable that is not in the probit equation. Otherwise, the model is identified only by functional form, and the coefficients have no structural interpretation [23]. Additionally, svyset command was used to adjust the complex design of the survey, which includes clustering and stratum effect. The analysis of the dataset has been carried out after assigning survey weight available in the data set. Moreover, Variance Inflation Factor (VIF) was estimated to check for multicollinearity [24], and no multicollinearity was found among the variables. Wald chi-square test was used to specify the goodness of fit for heckprobit model [23]. In STATA 14, we

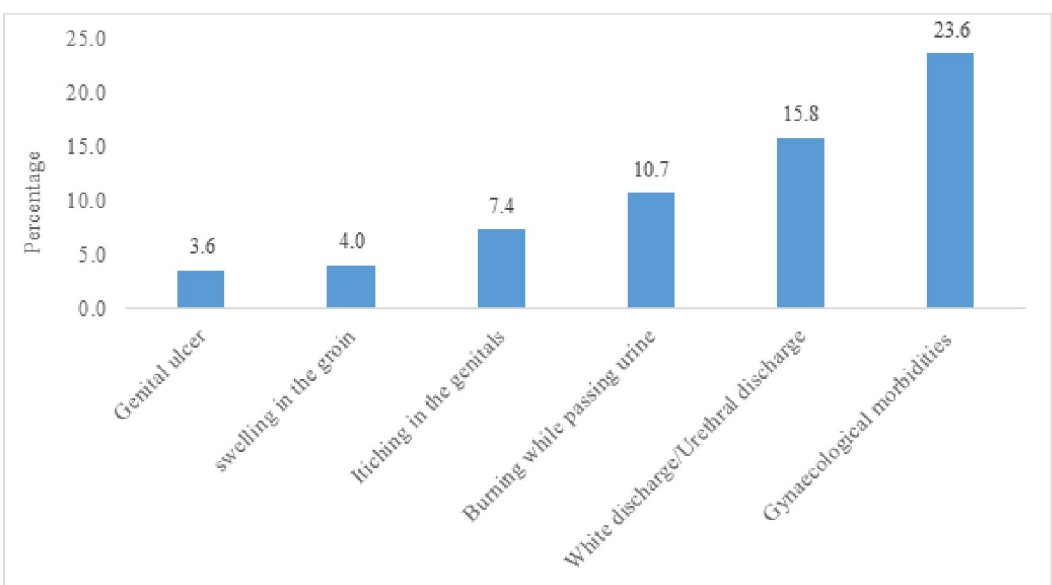

**Fig 1. Percentage of adolescent girls who reported gynaecological morbidities in three months prior to the survey year.**

used rvfplot command to check for heteroskedasticity, and it was found that there was no heteroskedasticity [25].

## Results

Fig 1 displays the different types of gynaecological morbidities among adolescents aged 10–19 years. About 16 percent of adolescents suffered from white discharge/urethral discharge, followed by burning while passing urine (10.7%) and itching in the genitals (7.4%).

The socio-demographic profile of adolescents aged 10–19 years was presented in Table 1. Around 37 percent of adolescents were sexually active, and half of the adolescents were used sanitary napkins. Interestingly, three-fifth of adolescents did not have toilet facilities, and most of them were 15–17 and 18–19 years age group. Nearly one-third of adolescents had ten and above years of education, 16.7 percent were working, and about half of the adolescents used media frequently. A higher proportion of adolescents were Hindu (78.5%) and belonged to rural areas (83.9%).

Gynaecological morbidities among adolescents and their treatment-seeking behavior were presented in Table 2. Overall, about one-fourth of the adolescents reported gynaecological morbidities, and one-third of them went for their treatment. Nearly one-third of sexually active adolescents suffered from gynaecological morbidities, and this was higher among adolescents who used sanitary napkins (26.8%). Interestingly, the gynaecological morbidities were significantly lower among adolescents who did not have toilet facilities than those who used toilet facilities. Gynaecological morbidities and their treatment-seeking behavior were positively associated with the age of adolescents. For instance, with an increase in the adolescents' age, the reporting of gynaecological morbidities and their treatment-seeking behavior also increased. Adolescents with no education (28.6%) reported more gynaecological morbidities, while adolescents with ten and above years of education (34.2%) went more for their treatment. Gynaecological morbidities were significantly higher among the working adolescents (27.4%) compared to those who were not working (22.9%). The reporting of gynaecological morbidities was higher among those who rarely had media exposure (26.3%), whereas

**Table 1. Demographic and socio-economic profile of adolescent girls aged 10–19 years.**

| Variables | Sample | Percentage |
|---|---|---|
| **Sexually active** | | |
| No | 9,213 | 63.0 |
| Yes | 5,412 | 37.0 |
| **Use of sanitary napkin** | | |
| Sanitary napkin | 7,307 | 50.0 |
| Cloth | 6,058 | 41.4 |
| Others | 1,260 | 8.6 |
| **Toilet facility** | | |
| Own flush/pit | 4,890 | 33.4 |
| Shared flush/pit | 1,022 | 7.0 |
| No facility | 8,712 | 59.6 |
| **Age (years)** | | |
| 10–12 | 944 | 6.5 |
| 13–14 | 709 | 4.9 |
| 15–17 | 6,780 | 46.4 |
| 18–19 | 6,192 | 42.3 |
| **Education (in years)** | | |
| No education | 1,890 | 12.9 |
| 1–7 | 3,939 | 26.9 |
| 8–9 | 4,093 | 28.0 |
| 10 and above | 4,703 | 32.2 |
| **Working status** | | |
| Not employed | 12,179 | 83.3 |
| Employed | 2,446 | 16.7 |
| **Media exposure** | | |
| No exposure | 2,703 | 18.5 |
| Rarely | 4,212 | 28.8 |
| Frequently | 7,710 | 52.7 |
| **Wealth index** | | |
| Poorest | 1,971 | 13.5 |
| Poorer | 2,735 | 18.7 |
| Middle | 3,188 | 21.8 |
| Richer | 3,577 | 24.5 |
| Richest | 3,154 | 21.6 |
| **Caste** | | |
| SC/ST | 3,784 | 25.9 |
| Non-SC/ST | 10,841 | 74.1 |
| **Religion** | | |
| Hindu | 11,479 | 78.5 |
| Non-Hindu | 3,146 | 21.5 |
| **Residence** | | |
| Urban | 2,356 | 16.1 |
| Rural | 12,269 | 83.9 |
| **State** | | |
| Uttar Pradesh | 9,855 | 67.4 |
| Bihar | 4,770 | 32.6 |

(*Continued*)

**Table 1.** (Continued)

| Variables | Sample | Percentage |
|-----------|--------|------------|
| Total | 14,625 | 100.0 |

SC/ST: Scheduled Caste/Scheduled Tribe

treatment-seeking for that was more among those who have frequently used mass media (35%). As expected, the richest adolescents (37.2%) went more for treatment than the poorest ones (20.7%). Adolescents who belonged to SC/ST group (24.9%) reported significantly higher gynaecological morbidities than non-SC/ST ones (23.2%). Moreover, this result was the opposite for treatment-seeking for gynaecological morbidities (25.4% vs. 34.2%). A higher proportion of adolescents belonging to urban areas (35.5%) seek treatment for gynecological morbidities than their rural counterparts (31.1%).

Estimates from the heck probit model for gynaecological morbidities and its treatment-seeking behavior among adolescents were presented in Table 3. The model was fit as the Wald chi-square test's value was statistically significant (65.24; p<0.05). Sexually active adolescents were 0.38 times more likely to suffer from gynaecological morbidities (β: 0.38; CI: 0.32–0.44) than those who were not sexually active. Gynaecological morbidities were 0.10 and 0.38 times significantly less likely among adolescents who used cloth and other materials, respectively, compared to those who used sanitary napkins. Adolescents age 15–17 (β: 0.28; CI: 0.09, 0.47) and 18–19 years (β: 0.36; CI: 0.17, 0.56) were 0.28 and 0.36 times more likely to have a gynaecological morbidities, respectively than adolescents with 10–12 years age group. Moreover, adolescents who belonged to the 15–17 and 18–19 years age group were 0.53 and 0.47 times less likely to go for the treatment of gynaecological morbidities, respectively, compared to 10–12 years adolescents. On the other hand, adolescents who had 8–9 standard (β: 0.17; CI: 0.05, 0.29) or ten and above (β: 0.21; CI: 0.09, 0.34) education were significantly 0.17 and 0.21 times more likely to go for treatment than illiterate ones. Non-SC/ST adolescents were 0.12 times significantly less likely to have gynaecological morbidities (β: -0.12; CI: -0.18, -0.06) compared to SC/ST counterparts. However, the same adolescent group was 0.09 times more likely to treat gynaecological morbidities (β: 0.09; CI: 0.00, 0.19). Moreover, adolescents who belonged to rural areas were 0.09 times less likely to go for the treatment of gynaecological morbidities (β: -0.09; CI: -0.17, -0.01) than urban counterparts.

## Discussion

This study examines gynaecological morbidities among adolescent girls aged 10–19 years and subsequent treatment for those gynaecological morbidities. The results from this study corroborate with previously available literature concerning risk factors for self-reported gynaecological morbidities and subsequent treatment for these morbidities. To say, our finding of increased risk of gynaecological morbidities among sexually active adolescent girls has been reported in various previous studies [26, 27]. Similarly, as in our study, various studies have reported a strong association between the use of shared toilets and the high prevalence of gynaecological morbidities among adolescent girls [28]. Further, the marked association between increasing age among adolescents and higher gynaecological morbidities is also logically documented in previous studies [29]. The study has several other significant findings. Gynaecological morbidities were higher among working adolescents, SC/ST adolescents, Non-Hindu adolescents, and adolescents in Uttar Pradesh.

**Table 2. Prevalence of gynaecological morbidity and its treatment-seeking behavior among adolescent girls aged 10–19 years by demographic and socio-economic characteristics.**

| Variables | Gynaecological morbidity (%) N = 14,625 | p<0.05 | Treatment seeking (%) N = 3,186 | p<0.05 |
|---|---|---|---|---|
| **Sexually active** | | * | | |
| No | 17.7 | | | |
| Yes | 33.8 | | | |
| **Use of sanitary napkin** | | * | | |
| Sanitary napkin | 26.8 | | | |
| Cloth | 23.3 | | | |
| Others | 6.9 | | | |
| **Toilet facility** | | * | | |
| Own flush/pit | 25.4 | | | |
| Shared flush/pit | 26.3 | | | |
| No facility | 22.4 | | | |
| **Age (years)** | | * | | * |
| 10–12 | 6.6 | | 19.2 | |
| 13–14 | 11.1 | | 21.5 | |
| 15–17 | 21.4 | | 25.3 | |
| 18–19 | 30.2 | | 37.6 | |
| **Education (in years)** | | * | | * |
| No education | 28.6 | | 29.2 | |
| 1–7 | 20.1 | | 30.5 | |
| 8–9 | 22.9 | | 31.2 | |
| 10 and above | 25.3 | | 34.2 | |
| **Working status** | | * | | |
| Not employed | 22.9 | | 32.6 | |
| Employed | 27.4 | | 28.5 | |
| **Media exposure** | | * | | * |
| No exposure | 21.4 | | 30.0 | |
| Rarely | 26.3 | | 27.6 | |
| Frequently | 23.0 | | 35.0 | |
| **Wealth index** | | | | * |
| Poorest | 21.2 | | 20.7 | |
| Poorer | 21.8 | | 29.6 | |
| Middle | 23.9 | | 34.0 | |
| Richer | 23.6 | | 31.5 | |
| Richest | 26.6 | | 37.2 | |
| **Caste** | | * | | * |
| SC/ST | 24.9 | | 25.4 | |
| Non-SC/ST | 23.2 | | 34.2 | |
| **Religion** | | * | | * |
| Hindu | 22.7 | | 29.4 | |
| Non-Hindu | 27.0 | | 39.2 | |
| **Residence** | | | | * |
| Urban | 23.5 | | 35.5 | |
| Rural | 23.7 | | 31.1 | |
| **State** | | * | | * |
| Uttar Pradesh | 25.7 | | 33.8 | |
| Bihar | 19.4 | | 26.1 | |

*(Continued)*

**Table 2.** (Continued)

| Variables | Gynaecological morbidity (%) N = 14,625 | p<0.05 | Treatment seeking (%) N = 3,186 | p<0.05 |
|---|---|---|---|---|
| Total | 23.6 | | 31.8 | |

%: percentage; SC/ST: Scheduled Caste/Scheduled Tribe;

*if p<0.05

Furthermore, the treatment for gynaecological morbidities was higher among educated adolescents, Non-SC/ST adolescents, adolescents in the urban area. The study noted that around one-fourth of the adolescent girls (23.6%) reported any one of the five gynaecological morbidities. Genital ulcer was the least reported, and white discharge/urethral discharge was reported by around 16 percent of the adolescents. As found in this study, the prevalence of various gynaecological morbidities was nearly the same as measured in previous studies in different settings in India [30–33].

Sexual activeness was found to be highly associated with gynaecological morbidities among adolescents. Previous studies also noticed a high level of gynaecological morbidities among sexually active [34, 35]. This study deviates from previous studies in noticing the positive association between the use of sanitary napkins and a low level of gynaecological morbidities [36, 37]. We are not sure about the mechanism of how this association was generated as we could not find any relevant literature; however, it could be presumed that the accumulation of blood in the genital area for a prolonged period may be a risk factor. For reasons like the high cost of the sanitary napkin, an adolescent girl may keep using the sanitary napkin for a longer duration than recommended; for these reasons, the association in our study was other way. A study in the Kenyan setting also noticed various factors associated with the use of sanitary napkin for a longer duration and assumed that using sanitary napkin for a longer duration may be a reason for the accumulation of blood in the genital area, which may further impact gynaecological morbidities [37]. The gynaecological morbidities were higher among adolescents who shared toilets than those adolescents who did not share toilets. A study highlighted higher gynaecological morbidities for those sharing toilets than those who do not share the toilet [36]. Sharing toilet seats may be a factor associated with high gynaecological morbidities [38].

Increasing age is one of the factors that was found to be associated with higher gynaecological morbidities among adolescents. Dheresa et al., in their systematic review, also noticed the association between age and gynaecological morbidities [29]. With an increase in age, adolescent girls may come across many risk factors of gynaecological morbidities, such as the onset of sexual life that may define higher gynaecological morbidities. Moreover, undiagnosed gynaecological morbidities at an earlier age may be diagnosed at a later age, thus, raising the prevalence of gynaecological morbidities at later ages. The finding that health-seeking for gynaecological morbidities declines with an increase in age is opposite to what was noticed by Savarkar in his study [39]. Therefore, an increase in age signifies the higher maturity level shall be attributed to the higher treatment-seeking for gynaecological morbidities.

Previous studies have highlighted the importance of education in declining the gynaecological morbidities among adolescents [29]. However, this study failed to find any significant association between education and gynaecological morbidities. Higher education, preferably, leads to lower reporting of gynaecological morbidities, probably because educated girls have a better knowledge of menstrual health, thereby reducing the chances of gynaecological morbidities [40]. Despite failing to associate education and gynaecological morbidities among adolescents, the study significantly concluded that treatment-seeking for gynaecological morbidities was higher among educated adolescents than their counterparts. Scholars unanimously have

**Table 3. Estimates from heck probit model for determinants of gynaecological morbidity and its treatment-seeking behavior among adolescents aged 10–19 years.**

| Background characteristics | Outcome equation | Selection Equation |
|---|---|---|
| **Sexually active** | | |
| No | Ref. | |
| Yes | 0.38*(0.32,0.44) | |
| **Use of sanitary napkin** | | |
| Sanitary napkin | Ref. | |
| Cloth | -0.10*(-0.15,-0.05) | |
| Others | -0.38*(-0.54,-0.22) | |
| **Toilet facility** | | |
| Own flush/pit | Ref. | |
| Shared flush/pit | 0.06* (0.02,0.14) | |
| No facility | -0.01(-0.07,0.06) | |
| **Age (years)** | | |
| 10–12 | Ref. | Ref. |
| 13–14 | 0.09(-0.11,0.29) | -0.50*(-0.84,-0.16) |
| 15–17 | 0.28*(0.09,0.47) | -0.53*(-0.79,-0.26) |
| 18–19 | 0.36*(0.17,0.56) | -0.47*(-0.75,-0.18) |
| **Education (in years)** | | |
| No education | Ref. | Ref. |
| 1–7 | 0.01(-0.07,0.09) | 0.04(-0.08,0.17) |
| 8–9 | -0.02(-0.1,0.06) | 0.17*(0.05,0.29) |
| 10 and above | -0.02(-0.1,0.07) | 0.21*(0.09,0.34) |
| **Working status** | | |
| Not employed | Ref. | Ref. |
| Employed | 0.21*(0.14,0.27) | -0.09(-0.19,0.01) |
| **Media exposure** | | |
| No exposure | Ref. | Ref. |
| Rarely | -0.21*(-0.33,-0.09) | 0.15*(0.08,0.23) |
| Frequently | -0.05(-0.18,0.07) | 0.06(-0.02,0.14) |
| **Wealth index** | | |
| Poorest | Ref. | Ref. |
| Poorer | -0.02(-0.11,0.07) | 0.12(-0.03,0.26) |
| Middle | 0.02(-0.07,0.11) | 0.11(-0.03,0.25) |
| Richer | 0.02(-0.08,0.11) | 0.12(-0.02,0.26) |
| Richest | 0.04(-0.06,0.15) | 0.09(-0.06,0.24) |
| **Caste** | | |
| SC/ST | Ref. | Ref. |
| Non-SC/ST | -0.12*(-0.18,-0.06) | 0.09*(0,0.19) |
| **Religion** | | |
| Hindu | Ref. | Ref. |
| Non-Hindu | 0.17*(0.11,0.24) | 0.03(-0.07,0.13) |
| **Residence** | | |
| Urban | Ref. | Ref. |
| Rural | 0.03(-0.03,0.08) | -0.09*(-0.17,-0.01) |
| **State** | | |
| Uttar Pradesh | Ref. | Ref. |
| Bihar | -0.19*(-0.24,-0.14) | -0.04(-0.13,0.04) |

*(Continued)*

**Table 3.** (Continued)

| Background characteristics | Outcome equation | Selection Equation |
|---|---|---|
| /athrho | -1.01*(-1.33,-0.7) | |
| rho | -0.77*(-0.87,-0.6) | |
| Wald chi2 | 65.24* | |
| Censored observation | 11,439 | |
| Uncensored observation | 3,186 | |
| Total observation | 14,625 | |

*P<0.05;

SC/ST: Scheduled Caste/Scheduled Tribe; Ref: Reference category

agreed on the association between higher education and higher levels of treatment-seeking for gynaecological morbidities [41]. Educated girls are well-informed about the consequences of gynaecological morbidities, and therefore, they seek treatment. The 'culture of silence' associated with gynaecological problems often hinders the participants from having an open discussion about their problems [42]. Females generally feel shy and disgrace to discuss the gynaecological problems with others [35]. Females often ignore the symptoms of gynaecological problems as these are perceived not so serious health issues [35]. 'Self-limiting' about the problem is the main reason for not seeking any healthcare [43].

Working status is another factor that was associated with gynaecological morbidities among adolescents in this study; however, the lower treatment-seeking for gynaecological morbidities among working adolescents was not significant in this study. Previous studies also highlighted that working women are more likely to suffer from gynaecological morbidities [44]. Working adolescents may find themselves busy with their work. Hence, personal hygiene and care may be left out because busy schedules could be a reason for high gynaecological morbidities among them.

Although a previous study noted that urban girls have better menstrual hygiene practices than rural girls [45], this study failed to find any association between reporting gynaecological morbidities among adolescent girls by their residence place. However, this study found that the treatment for gynaecological morbidities was lower among adolescent girls in rural areas than in urban areas. Previous studies align with our study in reporting lower treatment-seeking for gynaecological morbidities among rural girls [46]. In rural areas, stigma related to gynaecological morbidities may be one reason for the lower treatment of gynaecological morbidities among adolescents [47]. Moreover, in rural areas, health care services may be too far from home [47]. In rural areas, most married women and adolescent girls do not seek treatment as they did not feel that treatment was needed [46].

The study has several potential limitations. Foremost, gynaecological morbidities were self-reported by the respondents. Previously studies have noted differences between self-reporting of gynaecological morbidities and gynaecological morbidities diagnosed through clinical examination [48]. Therefore, we assume an underreporting of gynaecological morbidities in this study. However, this study measured gynaecological morbidities with a set of five questions, and therefore, the underreporting may not be to a greater extent. Another limitation is the period for which the gynaecological morbidities were recorded among respondents. Our study captured gynaecological morbidities for the past three months from the survey's date. The study sample covers only two states in India, and therefore the implications may differ from the wider population. Despite the limitations mentioned earlier, this study contributes to

a better understanding of gynaecological morbidities and their treatment-seeking among adolescents.

## Conclusion

Previously, several studies have examined menstrual hygiene among adolescents in various Indian settings; however, minimal scholarship exists for prevalence and factors associated with gynaecological morbidities and the subsequent treatment for gynaecological morbidities among adolescents. This study has several significant findings and has importance from a policy perspective. Addressing gynaecological morbidities among adolescent girls is a complex process as adolescents either do not consider it a significant health problem or hesitate to discuss it. Multi-pronged interventions are the need of the hour to raise awareness about the healthcare-seeking behaviour for gynaecological morbidities, especially in rural areas. Adolescent girls shall be prioritized as they may lack the knowledge for gynaecological morbidities, and such morbidities may go unnoticed for years. The mobile clinic may be the right approach as they have an educational outreach component too [49]. Mobile clinics may be used to disseminate appropriate knowledge among adolescents and screen asymptomatic adolescents for any possible gynaecological morbidity.

## Acknowledgments

The authors are thankful to David Jean Simon for copy editing the manuscript.

## Author Contributions

**Conceptualization:** Pradeep Kumar, Shobhit Srivastava, Shekhar Chauhan, Ratna Patel, Strong P. Marbaniang.

**Data curation:** Shobhit Srivastava.

**Formal analysis:** Pradeep Kumar, Shobhit Srivastava.

**Investigation:** Pradeep Kumar, Shobhit Srivastava, Shekhar Chauhan, Ratna Patel, Strong P. Marbaniang, Preeti Dhillon.

**Methodology:** Pradeep Kumar, Shobhit Srivastava.

**Software:** Shobhit Srivastava.

**Supervision:** Pradeep Kumar, Shekhar Chauhan, Ratna Patel, Strong P. Marbaniang, Preeti Dhillon.

**Validation:** Pradeep Kumar, Shobhit Srivastava, Shekhar Chauhan, Ratna Patel, Strong P. Marbaniang, Preeti Dhillon.

**Visualization:** Shobhit Srivastava, Preeti Dhillon.

**Writing – original draft:** Shekhar Chauhan, Ratna Patel, Strong P. Marbaniang.

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
