## [Decision Letter · Decision Letter 0]

15 Mar 2021

PONE-D-20-29552

Gynaecological morbidities and treatment-seeking among adolescent girls: A Heckprobit approach

PLOS ONE

Dear Dr. Marbaniang,

Thank you for submitting your manuscript to PLOS ONE. After careful consideration, we feel that it has merit but does not fully meet PLOS ONE’s publication criteria as it currently stands. Therefore, we invite you to submit a revised version of the manuscript that addresses the points raised during the review process.

3 expert reviewers revised the paper both in terms of data, methods and content. Their reports suggest that the paper requires substantial improvements in order to meet our publication requirements.

First, the language should be improved and the manuscript copy edited.

Second, regarding content, PLOS ONE endorses the STROBE initiative as a check of whether the research is appropriately carried out and reported. The article is currenty missing:

- A better placement in the literature on health-seeking behaviour as suggested by reviewer 1.

- A gap between the reported purpose and the research actually carried out as suggested by reviewer 1.

- A need to be precise about research objectives.

- Inadequate justification for the use of the Heckprobit model as commented by reviewers 2 and 3.

- Specific issues in model evaluation raised by the 3 reviewers.

- The discussion section needs to be improved in the light of a better introduction.

It is not necessary to skip the univariate analysis as suggested by reviewer 2. It helps the reader in acquiring familiarity with the data, and it is useful for assessing balance.

We look forward to receiving your revised manuscript.

Kind regards,

José Antonio Ortega, Ph.D.

Academic Editor

PLOS ONE

Journal Requirements:

2. Please list the name and version of any software package used for statistical analysis, alongside any relevant references.

For more information on PLOS ONE's expectations for statistical reporting, please see https://journals.plos.org/plosone/s/submission-guidelines.#loc-statistical-reporting

Reviewers' comments:

Reviewer's Responses to Questions

**Comments to the Author**

1. Is the manuscript technically sound, and do the data support the conclusions?

Reviewer #1: Partly

Reviewer #2: No

Reviewer #3: Partly

2. Has the statistical analysis been performed appropriately and rigorously? 

Reviewer #1: No

Reviewer #2: No

Reviewer #3: Yes

3. Have the authors made all data underlying the findings in their manuscript fully available?

Reviewer #1: Yes

Reviewer #2: No

Reviewer #3: Yes

4. Is the manuscript presented in an intelligible fashion and written in standard English?

Reviewer #1: Yes

Reviewer #2: Yes

Reviewer #3: No

5. Review Comments to the Author

Reviewer #1: This is an important area of research that will contribute to a growing evidence base. There are important concerns with the analysis, however, that require major revision.

Overall framing:

1. The authors should conduct a literature review focussed on the specific topic and population: they have not referred to key publications in this domain that focus on adolescents. The literature cited is predominantly about women of reproductive age, which is not appropriate given the specific factors that influence young women's treatment-seeking, and the separate issues amongst married and unmarried women. Some key papers (and there are more) include: Sabarwal and Santhya (2012) analysis of treatment-seeking amongst unmarried and married adolescent girls, using the Youth in India data; Jejeebhoy and Santhya (2011) review of SRH of young people in India; Sivakami's 2019 review of ARSH in India; Nagarkar’s systematic review on prevalence and treatment-seeking for RTI/STIs in India. Once the authors review the literature, they can be clear that the contribution of this paper is 1) a focus on adolescent girls 2) analysis of a range of factors associated with prevalence and treatment-seeking. Also, this sentence will no longer hold once they review the literature: "Previous studies have explored factors associated with treatment-seeking behaviour for gynaecological; however, failed to address the effect of women’s socio-economic factors."

2. The authors refer to gynecological morbidity throughout the paper, including menstrual disorders. The questions in the UDAYA survey, however, focus only on symptoms of genital infections (similar to the NFHS-4). The authors can highlight this difference, and ensure they are more specific in their use terms specifically on this sub-set of gynaecological morbidity, especially in the introduction and discussion.

3. They must refer to more recent policy in India specific to adolescents, especially the RKSK. The background actually focuses on adult women, which is not linked to the study or the analysis conducted by the authors.

Analysis:

1. The study is not designed to combine married and unmarried adolescents as a combined sample without applying appropriate weights. It is unclear whether the authors used these weights. Further, and more importantly, the literature suggests different factors contribute to gynaecological morbidity amongst unmarried and married adolescents. Accordingly, the authors should instead present findings disaggregated by marital status.

2. There are several more variables in the UDAYA study that could warrant inclusion in the analysis, such as awareness of SRH, discussion with parents, experience of violence. Please examine the full set of available variables carefully and provide an evidence-based justification for variables included, for both analyses.

3. The analysis of factors associated with treatment-seeking should consider a different set of variables than those associated with prevalence.There is a wide literature on treatment-seeking in India that can serve as a basis for inclusion.

4. The caste category should be split more finely, according to most analyses in this area.

5. Treatment-seeking descriptives can include description of where treatment was sought (and please see Sabarwal and Santhya analysis of treatment by sector amognst young women).

Reviewer #2: Thanks for the opportunity to review the methodological section of this manuscript. The manuscript as presently written is not methodologically sound and the findings does not warrant publication except the authors are willing to address the comments below:

1) Authors should change the title of the manuscript to "factors associated with treatment-seeking behaviour among adolescent girls residing in Bihar and Uttar Pradesh, India".

2) I guess the authors meant outcome variables rather than explanatory variables. Authors should change accordingly.

3) What is the theory underpinning this study? Authors should provide a theoretical framework for the study.

4) How was predictor variables selected? Authors need to state this in the manuscript and provide necessary references. Furthermore, work status cannot be yes or no. It is either they are employed or unemployed.

5) How was socio-economic status constructed? Authors would do well to explain how the SES index was constructed and how the quintiles were arrived at.

6) Authors need to provide the model specification for Heckman selection model and justify why they chose to use the model. It would be important to explain the issue of self-selection bias. See below:

Heckman JJ (1979) Sample selection bias as a specification error. Econometrica 47(1):153–161. https://doi.org/10.2307/1912352

7) The findings of this study cannot be relied upon as presently written. Authors need to address issues of multicollinearity, goodness of fit, endogeniety and heteroskedasticity in their analysis. A write-up on this issues would improve the findings of the study.

Reviewer #3: The study aimed to examine the prevalence and factors associated with gynaecological

morbidities and the treatment-seeking behaviour among adolescent girls residing in Bihar and

Uttar Pradesh, India

1. The grammar is quite bad, and needs huge improvements. For example, there are sentences like “Do the respondent had any gynaecological morbidities in the last three months? (Yes or no)”) on page 12. The paper needs to be worked on by a person fluent in the English language.

2. The rationale of using Heckman’s selection model seems mechanical and is not explained well; what is the selection issue and why this was needed.

3. Since the approach is statistical and regression results are being reported, I don’t think univariate results need to be discussed. Only summary statistics of the variables used is required. The entire idea of the Heckprobit regression is to control for other variables and report the probabilities.

4. I don’t understand Table 2. Are these percentages in total? For ex sexually active is 17.7 and not active is 33.8. Should these not add up to 1? Same is true of all other rows. These numbers should be revised and presented as background for those who had gynaecological morbidities.

5. Some explanations for the independent variables included need to be given.

6. What about mother’s education? Is that not an important variable for the sexual health of girls?

7. The reporting of results is not standard. In every result reported one does not have to write the CI and β values like this (β: 0.21; CI: 0.09, 0.34)

8. The summary and conclusion section should not repeat every result; instead make it rich by bringing in other evidence and explaining some results that are non-standard and specific to the Indian context and the possible reasons. These would be speculative I understand, but worth discussing.

Overall, with a good database this could turn into a neat paper, but need a lot more richness in discussions and policy implications, more use of intuitive explanations and huge improvements in style and language of writing.

6. PLOS authors have the option to publish the peer review history of their article (what does this mean?). If published, this will include your full peer review and any attached files.

Reviewer #1: No

Reviewer #2: **Yes: **Bolaji Samson Aregbeshola

Reviewer #3: No

---

## [Author Response · Author response to Decision Letter 0]

14 Apr 2021

Editor’s queries:

1. First, the language should be improved and the manuscript copy edited.

Response: The language of the manuscript has been improved. Also, we took help from one of the native English speakers for copy editing. Therefore, the acknowledgement section has been updated.

2. Second, regarding content, PLOS ONE endorses the STROBE initiative as a check of whether the research is appropriately carried out and reported. The article is currently missing:

Response: The article does not require the STROBE checklist as the manuscript used secondary source of data. However, it is reiterated that we have carried out the research by following the STROBE guidelines.

- A better placement in the literature on health-seeking behaviour as suggested by reviewer 1.

Response: The literature review has been revised as suggested.

- A gap between the reported purpose and the research actually carried out as suggested by reviewer 1.

Response: The raised query has been well-taken and accordingly it has been revised.

- A need to be precise about research objectives.

Response: Study Objective has been mentioned in the Introduction part precisely as suggested.

- Inadequate justification for the use of the Heckprobit model as commented by reviewers 2 and 3.

Response: Comment incorporated. 

- Specific issues in model evaluation raised by the 3 reviewers.

Response: Comment incorporated. 

- The discussion section needs to be improved in the light of a better introduction.

Response: The discussion section has been revised as per given suggestions.

It is not necessary to skip the univariate analysis as suggested by reviewer 2. It helps the reader in acquiring familiarity with the data, and it is useful for assessing balance.

Response: Table 1 is providing the univariate analysis.

Reviewer #1: 

This is an important area of research that will contribute to a growing evidence base. There are important concerns with the analysis, however, that require major revision.

Response: Thank you for your appreciation. We have carried out the revisions as suggested.

Overall framing:

1. The authors should conduct a literature review focussed on the specific topic and population: they have not referred to key publications in this domain that focus on adolescents. The literature cited is predominantly about women of reproductive age, which is not appropriate given the specific factors that influence young women's treatment-seeking, and the separate issues amongst married and unmarried women. Some key papers (and there are more) include: Sabarwal and Santhya (2012) analysis of treatment-seeking amongst unmarried and married adolescent girls, using the Youth in India data; Jejeebhoy and Santhya (2011) review of SRH of young people in India; Sivakami's 2019 review of ARSH in India; Nagarkar’s systematic review on prevalence and treatment-seeking for RTI/STIs in India. Once the authors review the literature, they can be clear that the contribution of this paper is 1) a focus on adolescent girls 2) analysis of a range of factors associated with prevalence and treatment-seeking. Also, this sentence will no longer hold once they review the literature: "Previous studies have explored factors associated with treatment-seeking behaviour for gynaecological; however, failed to address the effect of women’s socio-economic factors."

Response: Thanks for the comment. We now have incorporated the suggestion

2. The authors refer to gynecological morbidity throughout the paper, including menstrual disorders. The questions in the UDAYA survey, however, focus only on symptoms of genital infections (similar to the NFHS-4). The authors can highlight this difference, and ensure they are more specific in their use terms specifically on this sub-set of gynaecological morbidity, especially in the introduction and discussion.

Response: Thanks for the comment. In this study we used the term gynaecological morbidity to represent the self-reported symptoms of reproductive health problems. Suggestion incorporated. 

3. They must refer to more recent policy in India specific to adolescents, especially the RKSK. The background actually focuses on adult women, which is not linked to the study or the analysis conducted by the authors.

Response: Thanks for the comment. We have incorporate the suggestion.

Analysis:

1. The study is not designed to combine married and unmarried adolescents as a combined sample without applying appropriate weights. It is unclear whether the authors used these weights. Further, and more importantly, the literature suggests different factors contribute to gynaecological morbidity amongst unmarried and married adolescents. Accordingly, the authors should instead present findings disaggregated by marital status.

Response: Dear sir, thank you for the useful insight. Appropriate weights were used while doing the analysis to provide the reliable estimates. This is now mentioned in the method section. As we used the survey weights appropriately, there was no difference to estimate gynaecological morbidity between unmarried and married adolescents. 

2. There are several more variables in the UDAYA study that could warrant inclusion in the analysis, such as awareness of SRH, discussion with parents, experience of violence. Please examine the full set of available variables carefully and provide an evidence-based justification for variables included, for both analyses.

Response: Dear sir, I agree with your comment. However, the aim of the paper was limited to analyse the socio-economic factors which effect the treatment seeking behaviour for gynaecological morbidities among adolescent girls.

3. The analysis of factors associated with treatment-seeking should consider a different set of variables than those associated with prevalence. There is a wide literature on treatment-seeking in India that can serve as a basis for inclusion.

Response: Dear sir, I agree with the comment you raised. However, when we apply heckprobit model the pre-requisite is that the selection equation should have one variable less than the outcome equation. Moreover, the variable included in the analysis were selected after reviewing the wide literature on treatment-seeking in India.

4. The caste category should be split more finely, according to most analyses in this area.

Response: Dear sir the caste categories were split as SC/ST and non-SC/ST as in one of the category the sample was too low. Moreover, we defined SC/ST as a deprived social community. 

5. Treatment-seeking descriptives can include description of where treatment was sought (and please see Sabarwal and Santhya analysis of treatment by sector amognst young women).

Response: Comment incorporated. 

Reviewer #2: 

Thanks for the opportunity to review the methodological section of this manuscript. The manuscript as presently written is not methodologically sound and the findings does not warrant publication except the authors are willing to address the comments below:

1) Authors should change the title of the manuscript to "factors associated with treatment-seeking behaviour among adolescent girls residing in Bihar and Uttar Pradesh, India".

Response: The title has been modified as suggested by the reviewer.

2) I guess the authors meant outcome variables rather than explanatory variables. Authors should change accordingly.

Response: Thank you for your keen observation. Comment incorporated. 

3) What is the theory underpinning this study? Authors should provide a theoretical framework for the study.

Response: The literature review has been updated signifying the importance of the study and therefore, underlining the framework.

4) How was predictor variables selected? Authors need to state this in the manuscript and provide necessary references. Furthermore, work status cannot be yes or no. It is either they are employed or unemployed.

Response: The variables were selected after carrying out the rigorous literature review. The same has been highlighted in the revised manuscript.

5) How was socio-economic status constructed? Authors would do well to explain how the SES index was constructed and how the quintiles were arrived at.

Response: Comment incorporated. 

6) Authors need to provide the model specification for Heckman selection model and justify why they chose to use the model. It would be important to explain the issue of self-selection bias. See below:

Heckman JJ (1979) Sample selection bias as a specification error. Econometrica 47(1):153–161. https://doi.org/10.2307/1912352

Response: comment incorporated. 

7) The findings of this study cannot be relied upon as presently written. Authors need to address issues of multicollinearity, goodness of fit, endogeniety and heteroskedasticity in their analysis. A write-up on this issues would improve the findings of the study.

Response: Dear sir, thanks for the insight. Authors have incorporate the comment in the manuscript. 

Reviewer #3: 

The study aimed to examine the prevalence and factors associated with gynaecological

morbidities and the treatment-seeking behaviour among adolescent girls residing in Bihar and

Uttar Pradesh, India

1. The grammar is quite bad, and needs huge improvements. For example, there are sentences like “Do the respondent had any gynaecological morbidities in the last three months? (Yes or no)”) on page 12. The paper needs to be worked on by a person fluent in the English language.

Response: We took help from one of the native English speakers for copyediting the manuscript. Furthermore, each author has read the manuscript critically for any error. Also, the example provided by the reviewer - “Do the respondent had any gynaecological morbidities in the last three months? (Yes or no)” and other such texts were directly taken from the survey’s questionnaire and therefore, we feel that they should not be altered.

2. The rationale of using Heckman’s selection model seems mechanical and is not explained well; what is the selection issue and why this was needed.

Response: Comment incorporated as reviewer-2 raised the similar issue. 

3. Since the approach is statistical and regression results are being reported, I don’t think univariate results need to be discussed. Only summary statistics of the variables used is required. The entire idea of the Heckprobit regression is to control for other variables and report the probabilities.

Response: Most of the variables used in the study were categorical in nature therefore uni-variate analysis has been given, which provides the socio-demographic information of the respondents. Moreover, only key variables has been discussed in the Table 1.

4. I don’t understand Table 2. Are these percentages in total? For ex sexually active is 17.7 and not active is 33.8. Should these not add up to 1? Same is true of all other rows. These numbers should be revised and presented as background for those who had gynaecological morbidities.

Response: No, the percentages is not total. Authors have presented data for those only who reported gynaecological morbidity. Those who did not reported gynaecological morbidity is not presented in the table 2.

5. Some explanations for the independent variables included need to be given.

Response: Comment incorporated. 

6. What about mother’s education? Is that not an important variable for the sexual health of girls?

Response: I agree with your comment. However, the analysis includes married girls. Therefore, their mother’s education would not play a significant role if she lives with their in-laws. 

7. The reporting of results is not standard. In every result reported one does not have to write the CI and β values like this (β: 0.21; CI: 0.09, 0.34)

Response: Thanks for the suggestion. Comment has been incorporated.

8. The summary and conclusion section should not repeat every result; instead make it rich by bringing in other evidence and explaining some results that are non-standard and specific to the Indian context and the possible reasons. These would be speculative I understand, but worth discussing.

Response: The discussion section has been revised as per given suggestion.

---

## [Decision Letter · Decision Letter 1]

18 May 2021

Factors associated with gynaecological morbidities and treatment-seeking behaviour among adolescent girls residing in Bihar and Uttar Pradesh, India

PONE-D-20-29552R1

Dear Dr. Marbaniang,

We’re pleased to inform you that your manuscript has been judged scientifically suitable for publication and will be formally accepted for publication once it meets all outstanding technical requirements.

The three previous reviewers were invited and all of them accepted but, unfortunately, due to pandemic conditions two were unable to complete their review. The third reviewer recommends accept. In the opinion of the academic editor the manuscript has improved drastically and the issues raised by the two other reviewers have been addressed.

Within one week, you’ll receive an e-mail detailing any required amendments. When these have been addressed, you’ll receive a formal acceptance letter and your manuscript will be scheduled for publication.

Kind regards,

José Antonio Ortega, Ph.D.

Academic Editor

PLOS ONE

Additional Editor Comments (optional):

Reviewers' comments:

Reviewer's Responses to Questions

**Comments to the Author**

1. If the authors have adequately addressed your comments raised in a previous round of review and you feel that this manuscript is now acceptable for publication, you may indicate that here to bypass the “Comments to the Author” section, enter your conflict of interest statement in the “Confidential to Editor” section, and submit your "Accept" recommendation.

Reviewer #2: All comments have been addressed

2. Is the manuscript technically sound, and do the data support the conclusions?

Reviewer #2: Yes

3. Has the statistical analysis been performed appropriately and rigorously? 

Reviewer #2: Yes

4. Have the authors made all data underlying the findings in their manuscript fully available?

Reviewer #2: Yes

5. Is the manuscript presented in an intelligible fashion and written in standard English?

Reviewer #2: Yes

6. Review Comments to the Author

Reviewer #2: The authors have addressed the comments raised, hence, the manuscript is suitable for Publication. The reviewer has no further comments for the authors.

7. PLOS authors have the option to publish the peer review history of their article (what does this mean?). If published, this will include your full peer review and any attached files.

Reviewer #2: No

---

## [Editor Report · Acceptance letter]

26 May 2021

PONE-D-20-29552R1 

Factors associated with gynaecological morbidities and treatment-seeking behaviour among adolescent girls residing in Bihar and Uttar Pradesh, India 

Dear Dr. Marbaniang:

I'm pleased to inform you that your manuscript has been deemed suitable for publication in PLOS ONE. Congratulations! Your manuscript is now with our production department. 

Kind regards, 

on behalf of

Dr. José Antonio Ortega 

Academic Editor

PLOS ONE